# Cognitive Status and Outcomes of Older People in Orthopedic Rehabilitation? A Retrospective-Cohort Study

**DOI:** 10.3390/geriatrics5010014

**Published:** 2020-03-02

**Authors:** Carissa Bernal Carrillo, Christopher Barr, Stacey George

**Affiliations:** College of Nursing and Health Sciences, Flinders University, GPO BOX 2100, Adelaide 5001, Australia; Carissa.bernalcarrillo@gmail.com (C.B.C.); Chris.barr@flinders.edu.au (C.B.)

**Keywords:** cognitive impairment, discharge destination, elective surgery, fractured neck of femur, rehabilitation outcomes

## Abstract

Background: Cognitive function of older people is not routinely assessed in orthopedic rehabilitation, after elective and non-elective surgery. The aim of this study was to assess cognitive impairment and its impact on both length of stay and functional outcomes, of older people admitted to orthopedic rehabilitation. Methods: Retrospective audit, inclusion criteria: aged >65 years, orthopedic diagnosis, discharged from hospital. Results: 116 files were audited, mean age of 82.3 (SD = 7.5) years. Diagnostic groups: fractured neck of femur, (*n* = 44, 37.98%); elective surgery (*n* = 42, 36.21%); and other orthopedic conditions (*n* = 30, 25.86%). Overall 71.55% (*n* = 83) had cognitive impairment, with a median of mild cognitive impairment across all diagnoses. Both measures of cognition (MoCA/FIM Cognitive) were significantly associated with length of stay (*p* < 0.01), function (*p* < 0.05), and discharge destination (*p* = 0.01). Conclusions: A high percentage of older orthopedic patients in rehabilitation with both elective and non-elective diagnoses have cognitive impairment. Cognitive screening is recommended for all older orthopedic patients in rehabilitation, to inform an individualized rehabilitation plan to improve outcomes and length of stay. Further research is required to explore cognitive strategies to maximize rehabilitation outcomes in the geriatric orthopedic population.

## 1. Introduction

Australia has an increasing number of patients being admitted to their rehabilitation units with orthopedic conditions. In 2018 there were 60,963 patients admitted to rehabilitation for orthopedic surgeries and injuries [1], which was an increase from 2013, at 49,014 [2]. With such a high increasing number of orthopedic patients being admitted to rehabilitation units, it is important to ensure that both the expected length of stay targets and maximum functional outcomes are reached.

National benchmarks in Australia [1] of orthopedic patients in rehabilitation are between 11.3 (elective surgery) and 20.4 days (orthopedics fractures). On average, the age of people admitted to orthopedic rehabilitation units is 73.7 years [1]. With Australia’s increasing population of people over the age of 65 years, 19% from 2008–2013 [3], the demand for services such as orthopedic rehabilitation is expected to increase. Furthermore there is an increase in the age of patients being admitted to hospital, resulting in a growing number patients admitted with poorer cognition, including those experiencing acute confusion who demonstrate normal cognition prior to admission [4]. Factors that appear to impact most significantly on inpatient rehabilitation outcomes, such as discharge destination and functional outcomes, include cognition [5,6,7], increased age [6], and medical co-morbidities [8].

Of these three areas, which have been demonstrated to influence the length of stay of older people in orthopedic rehabilitation, the area which therapists can target to maximize function and improve outcomes, is cognition. Cognitive changes in rehabilitation may be pre-existing, which may be a factor leading to the person’s orthopedic injury, or occur acutely post-surgery, termed delirium, with resulting fluctuating changes in attention and cognition [4]. Rehabilitation from a therapy point of view may not specifically target the cognitive impairments but rather, through the design of the rehabilitation program, incorporate cognitive function to maximize engagement in rehabilitation, thereby improving functional outcomes. Evidence suggests that cognitively-impaired patients in the geriatric population can benefit from specialized rehabilitation programs, that is more intensive, with patients with moderate to severe cognitive impairment demonstrated to show functional improvements in rehabilitation [9]. What is needed is a better understanding of how, in orthopedic rehabilitation, a patient’s cognitive abilities can influence their rehabilitation process, and how rehabilitation can be delivered to maximize outcomes for people with cognitive changes [10]. It is suggested that patients with cognitive impairment may benefit from a rehabilitation program which is pragmatic, based on function, with repetition of routine daily tasks [9].

Many studies have been completed for patients’ post hip fracture including: randomized controlled trials on the effects of rehabilitation [11,12], comparative studies which review assessment tools to measure outcomes [5], and a review of the predictive factors of outcomes [6]. However, there is a dearth of literature that reviews the impact of cognition on the outcomes of other orthopedic conditions, including elective surgeries such as hip/knee replacements and upper/lower limb and pelvic fracture, in the geriatric population admitted to inpatient orthopedic rehabilitation.

Cognitive status can be an important predictor of rehabilitation outcomes [13], with having information related to a patient’s cognitive abilities and area of deficits assisting staff with providing retraining in functional task performance, including walking and activities of daily living. To maximize the functional outcomes of all geriatric orthopedic patients, not just those with hip fractures, a better understanding of a patient’s cognitive abilities and how this can influence their rehabilitation process is required.

Many rehabilitation units use cognitive screens to assist the decision-making processes about both a patient’s appropriateness to be admitted to rehabilitation, and timing and support required for discharge, however the screening tools used are not consistent [13]. In the literature, the main assessments reported include the Mini Mental State Examination (MMSE) [5] and the Montreal Cognitive Assessment (MoCA) [14]. Recent research investigating the association between older adults’ cognitive performance (using the MMSE and MoCA) and length of stay in a geriatric evaluation and management ward, found the orientation subscales of both tools were related to length of stay [15]. This was of small cohort (20 participants) with a range of conditions including deconditioning and falls. The MMSE, although widely used, has been shown to not be as effective in identifying mild cognitive deficits when compared to the MoCA [13]. The Functional Independence Measure (FIM) which is standardized functional measure used in rehabilitation units across Australia, is also widely used for measuring the outcomes of rehabilitation patients [6] and has a cognitive component. Both the MoCA and the cognitive component of the FIM, can be used to identify areas requiring treatment and intervention to maximize outcomes post rehabilitation.

Due to the relationship between cognition and function, it is important for rehabilitation teams to be aware of the cognitive status and areas of deficit of each individual patient. Patients with cognitive impairments show poorer functional gains on outcomes measures such as the Functional Independence Measure (FIM) [5,6,7] and Modified Barthel Index (MBI) [16]. Patients with cognitive impairment not only demonstrate deficits in the cognitive areas of the FIM, but also in the areas of self-care, mobility/transfers and bowel and bladder management in both the FIM and MBI. By identifying the areas of cognitive deficits, this information can then be used to retrain, cue, and prompt the patient appropriately and assist with decreasing carer burden, length of stay, and facilitating patients to return home.

The aim of this study was to investigate, through a retrospective cohort study, the relationship between the cognition, measured using the MoCA, and functional outcomes, measured using the FIM and MBI, of geriatric orthopedic patients admitted to rehabilitation, with diagnoses of fractured neck of femur, elective and other orthopedic surgery, and the length of their stay.

Occupational therapists provide intervention related to improving function in orthopedic rehabilitation. The occupational therapy intervention provided where the study occurred, was consistent across diagnoses, with rehabilitation intervention specifically targeting the needs of the individuals. All patients were seen by the occupational therapist for an initial assessment (including gaining information about their previous and current level of function, home environment, equipment, and modifications), goal setting, and a personal care assessment was generally completed within 72 h of admission (unless mobility was significantly impaired, that is requiring a hoist transfer, assisted by two people). As the patients’ mobility improved, functional assessments and retraining were completed, including a tea and toast assessment and functional retraining in breakfast and lunch groups. Patients on hip precautions also received dressing practice and education on the use of long handled aids. Patients also received functional mobility re-training, strength and endurance therapy, and in some cases cognitive-based therapy. Cognitive therapy was done using paper-based tasks or an iPad, with tasks set based on patients’ impairment. For example, if recall was the main impairment, therapy focused on recall tasks like memory. Patients, who also changed to a mobility aid that they were not previously using or if their home was not set up with aids or equipment, had a home visit completed. Rehabilitation input was not determined not by diagnosis but by the patient’s individual needs.

It was hypothesized that an association between cognitive impairment and outcomes in orthopedic rehabilitation would occur for patients admitted with fractures, who tend be older and more frail, compared to those admitted for elective procedures.

## 2. Materials and Methods

The audit was approved by the hospital’s Research and Ethics Committee, and participants identified through admission to the general rehabilitation unit at a metropolitan hospital in Brisbane, Queensland, Australia. The unit is staffed with a multi-disciplinary team, including physiotherapy, occupational therapy, dietetics, speech pathology, social work, nursing, and medical. Inclusion criteria was: patients over 65 years of age; admitted with the diagnoses of an orthopedic injury and surgeries (elective and non-elective), assessed using the MoCA [14] on admission, MBI [17] and FIM [18] on admission and discharge; and were discharged from the hospital. Data and information for this retrospective cohort study was collected via chart entries and admission and discharge information that is routinely collected by the occupational therapy team.

The three scales used were:MoCA [14]: This is a valid and reliable cognitive screen. The assessment is easy to administer with 18 questions broken down into the areas of visuospatial/executive, naming, memory, attention, language, abstraction, delayed recall, and orientation. The total score is out of 30, with scores being broken down into the following severity levels: intact (26–30), mild cognitive impairment (18–25), moderate cognitive impairment (10–17), and severe impairment (9 or less). The occupational therapist completing the assessments was trained to use the MoCA.MBI [17]: The MBI is a standardized assessment consisting of 10 areas: personal hygiene, bathing, feeding, toileting, stair climbing, dressing, bowel control, bladder control, ambulation and chair/bed transfers. Patients are scored from 1–5, with 1 indicating unable and 5, independent. Total score is out of 100, with a score of 100 indicating independent, 91–99 minimal assistance, 75–90 mild dependence, 50–74 moderate dependence, 25–49 severe dependence, and 0–24 total dependence. The MBI was also completed by the patient’s treating occupational therapist, completed on admission and discharge.FIM [18]: The FIM consists of a cognitive section with five areas assessed: comprehension, expression, social interaction, problem solving, and memory. Each area is scored out of 1–7, with 1 = total assistance and 7 = total independence. The total score for the cognitive area is 35, with a low score indicating increased assistance is required. The occupational therapist completing the cognitive section of the FIM had been FIM credentialed to be able to complete the assessment. The patients’ function was also assessed using the FIM motor section. The motor section is broken into 13 areas: eating, grooming, bathing, dressing upper body, dressing lower body, toileting, bladder management, bowel management, and transfers (bed, chair, wheelchair, transfer toilet, transfer shower, walk/wheelchair mobility, and stairs). This is also assessed from 1–7 with a total score out of 91, with a lower score indicating increased need for assistance. This motor section of the FIM was completed by the treating occupational therapist, physiotherapist and nurse who had been FIM credentialed to be able to complete the assessment.

Demographic and clinical information including age, length of stay, and diagnoses was also recorded. Diagnoses were grouped into sub-groups of: fractured neck of femur, elective orthopedic surgeries including hip and knee replacements, and other orthopedic injury/surgeries including orthopedic fractures of upper and lower limbs and fractured pelvis. Cognition was assessed on admission using the MoCA and patients grouped (into severity group as per the rating scale) to compare the difference between the diagnosis groups, with 26–30 within normal range, 18–25 mild cognitive impairment, 10–17 moderate cognitive impairment, and 9 or less severe cognitive impairment. 

Non-parametric statistics were used due to the skewness of the data, with all clinical characteristics of the subgroups of diagnoses presented as medians with inter-quartile ranges, and the relationship between diagnostic group and cognitive scores investigated using the Kruskal–Wallis test. The relationship between discharge location and MoCA scores was compared using the Mann–Whitney *U* test. Spearman’s correlation coefficients (r_s_) were used to examine the relationship between continuous scores, and chi-square tests (X^2^) for categorical data. All statistical analysis was conducted using SPSS version 23 [19].

## 3. Results

A total number of 152 patients were admitted to the rehabilitation unit from April–September 2014 following orthopedic injuries and surgeries (elective and non-elective). The number of patients included in this study was 116, with 36 excluded due to not meeting the inclusion criteria, (*n* = 15 due to the MoCA not being completed, *n* = 13 due to MBI not being completed, and *n* = 8 were under the age of 65 years).

Of 116 patients included in this study, *n* = 84 (72.4%) were females, with age ranging from 65 to 94 years, and mean age 82.30 (SD = 7.54) years. Diagnostic groups included: fractured neck of femur, (*n* = 44, 37.98%); elective surgery (*n* = 42, 36.20%); and other orthopedic conditions (*n* = 30, 25.86%). Elective surgeries included: total knee, total hip, and shoulder replacements which were pre-planned and scheduled. Orthopedic other included: upper (*n* = 4) and lower limb (*n* = 17) surgeries that were not planned, fractured pelvis, or patella fracture that did not fall under the other categories and there were not adequate numbers to make an individual sub-category. Patients’ clinical characteristics are presented in Table 1.

Clinical characteristics of the sub-groups based on diagnoses are presented in Table 2. The overall median MoCA score of each diagnosis was categorized as mild impairment. A Kruskal–Wallis test found there was no significant association between diagnostic groups and MoCA impairment scores (*p* = 0.106), nor using a chi-Square tests of categories (X^2^ = 6.19 (6) *p* = 0.35).

There was a significant difference between FIM (cognitive and motor), MBI scores on admission/discharge, and age, across diagnoses. Post hoc analysis of the significant difference between FIM cognitive scores at admission based on diagnosis, revealed that elective surgery patients’ scores were significantly higher than patients with fractured neck of femur (NOF) (*p* = 0.027). However, there was no difference between other orthopedic patients and fractured NOF or elective surgery patients. Similarly, the Kruskal–Wallis test also indicated a difference in FIM cognitive scores on discharge between the patients with fractured NOF and elective surgery patients (*p* = 0.46). There was again no significant difference between other orthopedic patients and fractured NOF or elective surgery patients.

Function was assessed using the FIM motor scores for admission and discharge and the MBI scores for admission and discharge. The Kruskal–Wallis test indicated a significant difference in FIM motor scores at admission based on diagnosis (*p* = 0.000). Post hoc analysis found that elective orthopedic patients’ scores were significantly higher than those of fracture NOF patients (*p* = 0.001) and other orthopedic patients (*p* = 0.000). There was no significant difference in FIM motor scores at admission for fractured NOF and other orthopedic patients. 

The Kruskal–Wallis Test also indicated a significant difference in FIM motor scores at discharge based on diagnosis (*p* = 0.000). Post hoc analysis revealed that elective orthopedic patients’ scores were significantly higher than those of fractured NOF patients (*p* = 0.003) and other orthopedic patients (*p* = 0.001). There was no difference in FIM motor scores at discharge for fractured NOF and other orthopedic patients. 

MBI admission and discharge scores were found to be significantly different when compared to diagnoses, with a significant association found, (*p* < 0.001). Post hoc analysis revealed the elective surgery patients’ scores were significantly higher than those of fractured NOF patients (*p* < 0.001) and other orthopedic patients (*p* < 0.001). There was no difference found between fractured NOF and other orthopedic patients. MBI discharge scores also indicated a significant difference, with this between the elective orthopedic patients and fractured NOF patients (*p* = 0.001). Analysis revealed no significant differences between MoCA scores and FIM motor or MBI changes (*p* > 0.05).

Refer to Table 3, for all associations between MoCA/FIM cognitive scores and functional outcomes. Length of stay was found to be significantly associated with diagnoses grouping. Post hoc analysis demonstrated a significant difference between the fractured NOF (median = 27) and orthopedic other (median = 31.5) group with a significantly longer length of stay, when compared to patients with elective orthopedic surgery (median = 13) (*p* < 0.001).

### Association Between MoCA/FIM Cognitive Scores and Functional Outcomes

FIM cognition scores and MoCA scores were compared to the functional variables using a Spearman’s Rho (r_s_). The associations between MoCA/FIM cognitive scores and functional outcomes are presented in Table 3. MoCA admission scores and FIM cognition and discharge scores were significantly associated with length of stay and FIM motor and MBI scores for the overall group.

The relationship between MoCA score and discharge location was examined using a Mann–Whitney *U* test. There was a significant difference (*p* = 0.010), with patients who were discharged home, who had a median score of 23 and those discharged to other locations (independent living unit, hostel, or nursing home) who had a median score of 19, with both scores indicating mild cognitive impairment. 

When the total group was divided into diagnoses for analyses the results were as follows.

Fractured NOF: There was not a significant association between MoCA or FIM cognitive scores on length of stay. There were significant associations between MoCA and admission/discharge FIM motor and MBI.

Elective Surgery: There was not a significant association between MoCA or FIM cognitive scores on length of stay. There were significant associations between FIM cognition admission/discharge with FIM motor admission/discharge and MBI admission. 

Other orthopedic: There was not a significant association between MoCA or FIM cognitive scores on length of stay. There was a significant association with MoCA scores in FIM motor admission and discharge and MBI discharge, and FIM cognitive admission/discharge with FIM motor (admission and discharge) and MBI (admission and discharge).

## 4. Discussion

This study indicates that a high number of geriatric orthopedic patients in rehabilitation, despite the reason for admission, that is elective and non-elective surgery, had cognitive impairment (71.55%, n = 83) as measured by the MoCA. This cohort demonstrated a median indicating mild cognitive impairment, with low scores on the MoCA, however a higher percentage of the fractured NOF group (80%) indicated cognitive impairment, with elective orthopedic surgery (60%) and orthopedic other (77%). 

There were no significant differences between the groups for their overall MoCA scores, however there was a significant difference between FIM cognitive scores on admission/discharge indicating that the functional cognition was lower for those admitted with a fractured NOF and other orthopedic conditions. With the range of diagnoses for the other orthopedic condition group and fractured NOF group, it could be assumed that these were due to falls for which pre-cognitive status may have contributed to the risk of falling. Interestingly the MBI was significantly different between the three diagnoses, indicating that the measures of function through observation, including FIM and MBI, examine differences in diagnostic groups. Interestingly the FIM motor changes for all groups were deemed clinically significant, which is >17 points [20], supporting its application as a functional outcome measure, on an orthopedic rehabilitation ward.

MBI change from admission to discharge was significantly different with the greatest improvement occurring in the other orthopedic group, closely followed by the fractured NOF group, supporting its use as an outcome measure of function in these patient populations. This also indicates that these patient had better functional outcomes, with similar cognitive abilities to those admitted for elective surgery. As has been noted in other studies patients admitted with fractures due to falls, commonly have declining function prior to admission [6,21] which influences functional outcomes, but may also provide more opportunity for improvement. Additionally, patients admitted for elective surgery may be better prepared prior to admission with social supports, home set-up and completing pre-admission exercises, and thus have a higher starting point of function, with less improvements measurable at the end of a rehabilitation program. Furthermore, the result that MoCA and change in FIM motor and MBI were not significant, does not support that people with worse cognition improve less in rehabilitation. However, a greater sample size with more variability of cognition is required.

Median length of stay for those patients admitted with a fractured NOF was 27 days and other orthopedic 31.5 days, whilst elective orthopedic surgery patients had a length of stay of 13 days. The Australasian Rehabilitation Outcomes Centre (AROC) [2], which is the national rehabilitation medicine clinical registry of Australia and New Zealand, reports the national benchmark for length of stay for elective orthopedic surgeries as follows: orthopedic fractures 21.9 days, elective surgery 11.8 days, and other orthopedic patients 13.9 days. When compared to the national bench marks [1] the patients in this study had an increased length of stay, and their functional level of dependence at discharge based on the MBI indicated mild dependence. This was beyond the scope of this project however may reflect different practices in rehabilitation or patient factors, such as high rates of cognitive impairments. It should also be noted that the average age of the patients admitted to this rehabilitation unit and reviewed for this study was higher than the national average. With the national average age for orthopedic patients being 73.7 years [1] and the average age of patients in this study being 82.30 years.

The findings indicate that all cognitive measures, that is both MoCA and FIM cognitive scores, were related to length of stay in the group overall but not in each diagnosis. This indicates that across both elective and non-elective orthopedic diagnoses, cognition can inform length of stay. With orthopedic rehabilitation demonstrated to increase hospital stays and subsequent costs [22], the relationship between cognition and length of stay indicates that cognition, even if minimal reductions are evident, needs to be targeted to minimize costs in healthcare services. This supports the recommendations of cognitive screens being used with all patients admitted to geriatric orthopedic rehabilitation units, despite diagnoses. This screening could assist programs to tailor how the rehabilitation is delivered, in terms of functional contexts, specific cuing, and the incorporation of cognitive training strategies to maximize functional outcomes and potentially reduce length of stay. Cognition is highly impacted by age and can influence the patient’s ability to actively participate in the rehabilitation process including goal setting, poor recall and learning, and difficulties with making decision about discharge location and services required [23,24].

The results indicate that for the fractured NOF group and the other orthopedic group the MoCA scores were related to the functional scores of FIM (admission and discharge) and MBI (discharge), thus indicating that the MoCA is useful in measuring cognition pre- and post-rehabilitation as related to functional measures at both points. However, for the elective surgery group the FIM cognitive, and not the MoCA, on admission was related to functional measures, that is FIM motor and MBI on admission, but only FIM motor on discharge. This suggests that for the elective surgery group the FIM cognitive is more useful in measuring the cognitive abilities that will relate to functional performance.

As there were no significant differences when comparing severity level on the MoCA to diagnoses, a further review of the breakdown of scores based on assessment areas (visuospatial/executive, naming, memory, attention, language, abstraction, delayed recall, and orientation) would be of benefit to determine which aspects of cognition relate to length of stay. Interestingly, MoCA scores and discharge location were significantly different between those who were discharged home and those discharged to other locations (independent living unit, hostel, or nursing home). Thus, even within the overall minimal cognitive impairment category of this group of geriatric orthopedic clients, cognitive scores provided an indication of destination for discharge planning.

Limitations of this study included that within the patients admitted to the orthopedic rehabilitation ward 15 were excluded as they did not complete the MoCA. Potentially these patients may have been more cognitively impaired than the included sample and therefore unable to complete the paper-based test of cognition. Thus, future prospective studies should consider utilizing other forms of cognitive screening that are observational in nature, such as the FIM cognitive. A further limitation is the use of a convenience sample of admissions across 6 months, however this was considered representative of the sample population. Logistic regression analysis should be considered in future studies with larger samples sizes. Although, retrospective study designs have inherent limitations the results highlight some important information about the relationship between cognition and orthopedic rehabilitation outcomes. Further limitations include no measures of the presence of delirium, which is transient cognitive impairment, nor measures of frailty. Frailty represents overall health and functional status, reducing the ability to overcome stressors, leading to adverse outcomes after surgery [25]. Both of these are now measured routinely in the service and were not at the time of data collection. In addition, comorbidities were not recorded, as they were not the primary area of investigation, but as they will impact on functional outcomes could be recorded in future studies.

A high proportion of the orthopedic population in this study, were measured as having cognitive impairment. With the evidence supporting potential functional gain in rehabilitation for those patients with moderate–severe cognitive impairment [15], further research into the ability to design rehabilitation programs to maximize functional outcomes for older people with cognitive changes is warranted. Details of specific cognitive deficits experienced by patients admitted to orthopedic wards would inform the design of rehabilitation programs. Additionally, with falls being the leading cause of orthopedic injury in older people, fall prevention programs, which include an education component, need to consider cognitive factors to maximize their effectiveness in reducing and preventing falls. Although cognitive retraining and intervention is typically undertaken with neurological patients, it is important to consider addressing the age-related deficits, particularly in the area of cognition with geriatric orthopedic patients to assist with improving functional outcomes and length of stay.

Experts support the consideration of how technology may improve outcomes and the facilitation of older people to direct their own rehabilitation process. Thus the application of technology to inform rehabilitation for people with cognitive changes in orthopedic rehabilitation [10] requires investigation. Further research needs to address the most effective way to provide rehabilitation to those older people with orthopedic conditions with cognitive impairment, who are mostly excluded from rehabilitation trials.

## 5. Conclusions

In conclusion, the geriatric orthopedic patients admitted to rehabilitation presented with cognitive impairment, across diagnoses and this was associated with discharge destination, functional outcomes, and length of stay. Cognitive screening is recommended for all geriatric orthopedic patients admitted to rehabilitation, to inform an individualized rehabilitation plan to improve outcomes and length of stay. Further research is required to explore cognitive strategies to maximize rehabilitation outcomes in the geriatric orthopedic population.

## Figures and Tables

**Table 1 geriatrics-05-00014-t001:** Clinical characteristics of patients.

	Mean (SD)
Rehabilitation length of stay	24.8 days (13.2)
MoCA	22 (5)
FIM	
Motor admission	56 (13)
Motor discharge	76 (11)
Cognitive admission	30 (5)
Cognitive discharge	31 (5)
MBI	
Admission	72 (16)
Discharge	90 (14)
	Number (%)
Diagnosis	
Fractured neck of femur (NOF)	44 (37.98%)
Elective surgeries	32 (27.58%)
Other orthopedic	40 (30.48%)

MoCA = Montreal Cognitive Assessment, FIM = Functional Independence Measure; MBI = Modified Barthel Index.

**Table 2 geriatrics-05-00014-t002:** Clinical characteristics based on diagnoses.

	Diagnosis			
Descriptive	Fractured NOF (N = 44)	Elective Orthopedic Surgery (N = 42)	Orthopedic Other (N = 30)	*p*-Value
Age (years) range, mean (SD)	66–94 84.96 (7.59)	65–92 79.59 (6.76)	65–91 82.20 (7.34)	0.001 *
Length of stay (days) Mean (SD)	27.69 (11.95)	15.90 (8.17)	33.23 (13.91)	<0.001*
**MoCA**	**(Median, IQR, Severity levels n%)**			
Median (IQR)	21 * (17–25)	24 *(20–26)	23 * (19–25)	0.106
Intact (n%)	9 (21)	13 (31)	7 (23)	
Mild(n%)	24 (55)	25 (60)	18 (60)	
Moderate (n%)	10 (22)	4 (9)	4 (13)	
Severe (n%)	1 (2)	0	1 (3)	
**FIM motor**	**Median (IQR)**			
Admission	50 (43–58)	63 (56–71)	52 (41–56)	<001 *
Discharge	79 (65–82)	83 (79–85)	77 (67–82)	<001 *
Change	24.5(14–31)	19 (11–25)	24 (10–30)	0.107
**FIM cognition**				
Admission	29 (26–33)	32 (29–35)	30 (26–35)	0.024 *
Discharge	30 (27–35)	34 (30–35)	32 (28–35)	0.043 *
Change	0 (0–2)	0 (0–1)	0 (0–2)	0.394
**MBI**	**Median (IQR) Dependence category (n%)**			
Admission (median, IQR)	69 (59–76)	85 (77–89)	71 (58–79)	<0.001*
Minimal assistance (n%)	0	0	1 (3)	
Mild dependence (n%)	14 (32)	8 (19)	9 (30)	
Moderate dependence (n%)	25 (57)	25 (60)	16 (53)	
Severe dependence (n%)	3 (7)	9 (21)	4 (13)	
Total dependence (n%)	2 (4)	0	0	
Discharge (median, IQR)	92 (81–96)	96 (94–99)	94 (84–98)	0.001*
Independent (n%)	2 (5)	10 (24)	3 (10)	
Minimal assistance (n%)	23 (52)	27 (64)	18 (60)	
Mild dependence (n%)	12 (27)	4 (10)	4(13)	
Moderate dependence (n%)	5 (12)	1 (2)	3 (10)	
Severe dependence (n%)	1 (2)	0	2 (7)	
Total dependence (n%)	1 (2)	0	0	
Change	20 (13–29)	12 (7–19)	22 (9–29)	0.005*

MoCA = Montreal Cognitive Assessment, FIM = Functional Independence Measure; MBI = Modified Barthel Index, **p* value significant at *p* < 0.05. *Indicating mild cognitive impairment.

**Table 3 geriatrics-05-00014-t003:** Association between MoCA/FIM cognitive scores and functional outcomes.

	MoCA Admission		FIM Cognition Admission		FIM Cognition Discharge	
	r_s_	*p*	r_s_	*p*	r_s_	*p*
All diagnoses						
Length of stay	−0.278	0.003 *	−0.300	0.001 **	−0.257	0.005 **
MoCA		-	0.559	0.001 **	0.563	0.001 **
FIM motor						
Admission	.385	0.001 **	0.509	0.001 **	0.460	0.001 *
Discharge	0.421	0.001 **	0.899	0.001 **	0.596	0.001 *
MBI						
Admission	0.390	0.001 **	0.486	0.001 **	0.430	0.001 *
Discharge	0.406	0.001 **	0.444	0.001 **	0.446	0.001 *
**Fractured NOF**						
Length of stay	−0.250	0.102	−0.251	0.101	−0.221	0.149
MoCA		-	0.590	0.001 **	0.674	0.0001 *
FIM motor						
Admission	0.333	0.027 *	0.483	0.001 **	0.443	0.003 **
Discharge	0.490	0.001 **	0.491	0.00 1**	0.563	0.000 **
MBI						
Admission	0.395	0.008 *	0.448	0.002 **	0.459	0.002 **
Discharge	0.490	0.001 **	0.514	0.000 **	0.537	0.000 **
**Elective Surgery**						
Length of stay	−0.2666	0.089	−0.186	0.239	−0.107	0.500
MoCA		-	0.456	0.0002 **	0.300	0.054
FIM motor						
Admission	0.151	0.340	0.406	0.008 **	0.325	0.036 *
Discharge	−0.117	0.461	0.327	0.035 *	0.389	0.011 *
MBI						
Admission	0.310	0.046	0.392	0.010 *	0.291	0.061
Discharge	0.064	0.688	0.217	0.168	0.167	0.290
**Other orthopedic**						
Length of stay	−0.137	0.469	−0.255	0.173	−0.208	0.270
MoCA		-	0.586	0.001**	0.636	0.000 **
FIM motor						
Admission	0.526	0.003 **	0.455	0.012*	0.428	0.019 *
Discharge	0.679	0.000 **	0.618	0.000**	0.697	0.000 **
MBI						
Admission	0.343	0.064	0.469	0.009 **	0.416	0.022 *
Discharge	0.606	0.000 **	0.485	0.007 **	0.576	0.001 *

MoCA = Montreal Cognitive Assessment, FIM = Functional Independence Measure; MBI = Modified Barthel Index, (r_s_) = Spearman’s correlation coefficients, * significant *p* < 0.05, ** significant *p* < 0.01 (2-tailed).

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
