# Peer review of "Cognitive Status and Outcomes of Older People in Orthopedic Rehabilitation? A Retrospective-Cohort Study"

_geriatrics, 2020, doi:10.3390/geriatrics5010014_

Round 1
Reviewer 1 Report
Lines 113-114: I am not sure that I understand this sentence. How can rehabilitation NOT be influenced by the patient diagnosis? Given the variability of diagnostic groups. how can all patients receive the same rehabilitation?
Lines 124-125: This sentence is somewhat inconsistent with lines 113-114. So therapy varied by therapist but rehabilitation did not vary based on diagnosis?
Results 187: I think that you need more descriptive data on what types of surgeries fall under "elective orthopedic surgeries" and "orthopedic other." You examine a specific type of surgery (status post fracture of the neck of the femur) and two other groups both of which are non-specific in diagnosis.
Lines 188-189: Why was this 6-month time period selected? Why did you pick a 6-month window that is from 5+ years ago?
Line 198: the "and" should be move before the "lower limb other..."
Line 212: what does the "(6): refer to inside the brackets?
Lines 329-330: does AROC data on elective surgery refer to "elective orthopedic surgery"? If so, please specify. Similar to the comments on line 187, other countries do not report or use data on these broad types of surgical categories, so you may provide some additional clarification for the readers.
Line 376:do you mean "within the patients included, 15 ..." Included would make sense, excluded does not make sense.
Lines 381-382: On what basis do you consider the 6-month sample used, which is 5+ years old, representative of the sample population? There is no information in your methods to suggest that the sample used was anything other than one of convenience.
Author Response
Dear Reviewer,
Thank you for your considered reviews. Please see responses below.
Reviewer 1
Lines 113-114: I am not sure that I understand this sentence. How can rehabilitation NOT be influenced by the patient diagnosis? Given the variability of diagnostic groups. how can all patients receive the same rehabilitation?
This has been adjusted to “Occupational Therapy process where the study occurred, was consistent across diagnoses, with rehabilitation intervention specifically targeting the needs of the individuals.” Lines 121- 123
Lines 124-125: This sentence is somewhat inconsistent with lines 113-114. So therapy varied by therapist but rehabilitation did not vary based on diagnosis?
This is now consistent with changes in Lines 113-114.
Results 187: I think that you need more descriptive data on what types of surgeries fall under "elective orthopedic surgeries" and "orthopedic other." You examine a specific type of surgery (status post fracture of the neck of the femur) and two other groups both of which are non-specific in diagnosis.
Details have been added Lines 207-212. Elective surgeries included: total knee, total hip replacements and shoulder replacements which were pre-planned and scheduled. Orthopedic other included: upper (n = 4) and lower limb (n = 17) surgeries that weren’t planned, fractured pelvis, or patella fracture that didn’t fall under the other categories and there were not adequate numbers to make an individual sub-category.
Lines 188-189: Why was this 6-month time period selected? Why did you pick a 6-month window that is from 5+ years ago?
The study was completed as part of a Masters project. At the time this was the most recent time period. 6 months was felt to allow for a good number of patients to be captured.
Line 198: the "and" should be move before the "lower limb other..." This has been deleted see Line 212
Line 212: what does the "(6): refer to inside the brackets? Degrees of freedom
Lines 329-330: does AROC data on elective surgery refer to "elective orthopedic surgery"? If so, please specify. Similar to the comments on line 187, other countries do not report or use data on these broad types of surgical categories, so you may provide some additional clarification for the readers.
Yes the data is on elective orthopaedic surgeries. This has been clarified Lines 342-345
Line 376:do you mean "within the patients included, 15 ..." Included would make sense, excluded does not make sense.
It is excluded as they were not included in the sample for analysis as the cognitive scores were not recorded as not measured. Clarity has been increased to: Limitations of this study include that within the patients admitted to the orthopedic rehabilitation ward 15 were excluded as they did not complete the MoCA. Lines 402-403
Lines 381-382: On what basis do you consider the 6-month sample used, which is 5+ years old, representative of the sample population? There is no information in your methods to suggest that the sample used was anything other than one of convenience.
This was done as part of masters research project and this was a convenience sample to answer the research question of what is the cognitive status of older people admitted to orthopaedic rehabilitation. This is addressed in the limitations Lines 398-400
The purpose of this retrospective study was to evaluate the association between cognitive impairment, length of stay and functional outcomes among older adults admitted to an inpatient rehabilitation unit following orthopaedic surgery. The authors posit that rehabilitation strategies should be tailored to level of cognitive impairment in order to reduce length of stay and maximize outcomes. This is not only important for older adults receiving surgery for a hip fracture, a common orthopedic injury, but also for other orthopaedic injuries and elective procedures as these too are increasing as the population ages. This paper highlights an important issue with clear clinical implications. However, the manuscript was difficult to follow and could be improved by addressing the following items:
Thank you for the above comments
Reviewer 2 Report
The purpose of this retrospective study was to evaluate the association between cognitive impairment, length of stay and functional outcomes among older adults admitted to an inpatient rehabilitation unit following orthopaedic surgery. The authors posit that rehabilitation strategies should be tailored to level of cognitive impairment in order to reduce length of stay and maximize outcomes. This is not only important for older adults receiving surgery for a hip fracture, a common orthopedic injury, but also for other orthopaedic injuries and elective procedures as these too are increasing as the population ages. This paper highlights an important issue with clear clinical implications. However, the manuscript was difficult to follow and could be improved by addressing the following items:
The introduction talks about falls but it is unclear how falls factor into the current analysis. Falls are the leading cause of orthopaedic injury in older adults, but the analysis presumably includes patients who did not fall (i.e the elective procedures). Since it seems that one of the important aspects for improving rehab care for people with cognitive impairment is to understand the areas of deficit, it would have been nice to see more descriptive information about the subscores of the MoCA and FIM cog. Did the authors have any hypotheses with regard to the association between cognitive impairment and outcomes across the three diagnosis groups (i.e. hip fracture, elective procedure, other fracture?) In the presentation of results in Table 2, it would have been helpful to provide the distribution by cognitive impairment and function category in addition to mean and IQR. For example, report % of those with intact, mild, moderate, severe CI; and independent, minimal assistance, mild dependent, etc. for the MBI. Also consider including the median values as these are reported in the text and was a bit confusing when referring back to the table. Please go carefully through the tables and text to correct number typos (example: table 2 FIM motor change for the hip fracture group says 255- I don’t thin that’s correct; page 9 line 262 “other orthopaedic” median looks incorrect). Page 10 line 316 indicates that the MBI change from admission to discharge was significantly different with the greatest improvement occurring in the elective surgery group. However, according to Table 2, the largest change is in the other orthopaedic group. When discussing differences across groups, it would be useful to know the clinically important differences in scores if one exists (see Beninato, 2006) If the idea is that cognition negatively impacts the effectiveness of rehab, you might look at the association between MoCA and change in FIM motor and MBI. Looks like on average, everyone regardless of diagnosis group improved from admission to discharge, but it’s possible that people with worse cognition improve less which would support the need for different strategies. May be some issues with sample size and limited of variability in cognitive impairment. In the discussion it’s worth noting that cognitive impairment may impact the rehab outcomes and goals of these patient groups differently. For example, and as was touched on in the discussion, patients with hip fracture are likely already on a decline when admitted to the hospital versus older adults that elect to have joint replacement surgery. There might also be some selection bias here by virtue of the nature of these two surgeries. Consider tempering the language regarding the association between MoCA and length of stay. While it was statistically significant when pooling data across groups, the correlation coefficient indicates a weak association. There are likely many other factors that contribute to length of stay. May be more appropriate to say that screening should be done at time of admission in order to create a rehab plan that accounts for cognitive impairment which would then improve outcomes sooner thereby reducing length of stay. Is it possible to extend the retrospective review to include more patients?Author Response
Dear Reviewer,
Thank you for your considered reviews. Please see responses below.
Reviewer 2The introduction talks about falls but it is unclear how falls factor into the current analysis. Falls are the leading cause of orthopaedic injury in older adults, but the analysis presumably includes patients who did not fall (i.e the elective procedures).
The aim of the study was to look at cognitive function more broadly across all older patients
admitted to an orthopaedic rehabilitation wards. Also comment added in the discussion Lines 415 to 417 Additionally, with falls being the leading cause of orthopedic injury in older people, falls prevention programs need to consider cognitive factors to maximize their effectiveness in reducing and preventing falls.
Since it seems that one of the important aspects for improving rehab care for people with cognitive impairment is to understand the areas of deficit, it would have been nice to see more descriptive information about the subscores of the MoCA and FIM cog.
This has been added to further research recommendations as is important but beyond the scope of the study. Lines 415-416 Details of specific cognitive deficits experienced by patients admitted to orthopedic wards would inform the design of rehabilitation programs.
Did the authors have any hypotheses with regard to the association between cognitive impairment and outcomes across the three diagnosis groups (i.e. hip fracture, elective procedure, other fracture?)
The following has been added Lines 141-143. It was hypothesized that an association between cognitive impairment and outcomes would occur for patients admitted with fractures, who tend be older and more frail, compared to those admitted for elective procedures.
In the presentation of results in Table 2, it would have been helpful to provide the distribution by cognitive impairment and function category in addition to mean and IQR. For example, report % of those with intact, mild, moderate, severe CI; and independent, minimal assistance, mild dependent, etc. for the MBI. Also consider including the median values as these are reported in the text and was a bit confusing when referring back to the table.
This has been completed
Please go carefully through the tables and text to correct number typos (example: table 2 FIM motor change for the hip fracture group says 255- I don’t think that’s correct; page 9 line 262 “other orthopaedic” median looks incorrect).
This has been checked and corrected
Page 10 line 316 indicates that the MBI change from admission to discharge was significantly different with the greatest improvement occurring in the elective surgery group. However, according to Table 2, the largest change is in the other orthopaedic group.
This has been checked and completed
When discussing differences across groups, it would be useful to know the clinically important differences in scores if one exists (see Beninato, 2006) If the idea is that cognition negatively impacts the effectiveness of rehab, you might look at the association between MoCA and change in FIM motor and MBI. Looks like on average, everyone regardless of diagnosis group improved from admission to discharge, but it’s possible that people with worse cognition improve less which would support the need for different strategies.
This has been considered in the discussion. No sig differences were found between Moca and functional outcomes- reported in results
May be some issues with sample size and limited of variability in cognitive impairment. In the discussion it’s worth noting that cognitive impairment may impact the rehab outcomes and goals of these patient groups differently. For example, and as was touched on in the discussion, patients with hip fracture are likely already on a decline when admitted to the hospital versus older adults that elect to have joint replacement surgery.
Included in discussion
There might also be some selection bias here by virtue of the nature of these two surgeries. Consider tempering the language regarding the association between MoCA and length of stay.
Included in limitations
While it was statistically significant when pooling data across groups, the correlation coefficient indicates a weak association. There are likely many other factors that contribute to length of stay. May be more appropriate to say that screening should be done at time of admission in order to create a rehab plan that accounts for cognitive impairment which would then improve outcomes sooner thereby reducing length of stay. This has been adjusted
Is it possible to extend the retrospective review to include more patients? Unfortunately no as a masters project but included to increase sample size in future research recommendations
Round 2
Reviewer 2 Report
Thank you for cleaning up the tables and data inconsistencies in the text and for addressing the additional items in your discussion. The beginning of the introduction is still framed around falls. If the intention was to look at the effect of cognitive impairment on rehabilitation outcomes in a broader population, the introduction should focus on the increase in orthopaedic injuries, and elective orthopaedic procedures. Otherwise, I feel it is misleading
Author Response
Thank you for your review. The introduction has been reworked to only focus on the increase of orthopedic injuries and elective surgery. References to falls have been removed.